# Stereotactic Radiotherapy in the Treatment of Paraneoplastic Vasculitis in Oligometastatic Renal Cell Carcinoma

Laura Burgess [1,*], Marissa Keenan [2], Alan Liang Zhou [3], Kiefer Lypka [3], Delvina Hasimja Saraqini [3], Jeff Yao [4], Samuel Martin [5], Christopher Morash [4], James Watterson [4], Christina Canil [6] and Robert MacRae [1]

[1] Department of Radiology, Division of Radiation Oncology, University of Ottawa, 501 Smyth Road, Ottawa, ON K1H8L6, Canada; rmacrae@toh.ca

[2] Department of Medicine, Division of Rheumatology, University of Ottawa, 501 Smyth Road, Ottawa, ON K1H8L6, Canada; mkeenan@toh.ca

[3] Department of Medicine, University of Ottawa, 501 Smyth Road, Ottawa, ON K1H8L6, Canada; alzhou@toh.ca (A.L.Z.); klypka@toh.ca (K.L.); dhasimja@toh.ca (D.H.S.)

[4] Department of Surgery, Division of Urology, University of Ottawa, 501 Smyth Road, Ottawa, ON K1H8L6, Canada; jeyao@toh.ca (J.Y.); cmorash@toh.ca (C.M.); jwatterson@toh.ca (J.W.)

[5] Faculty of Medicine, University of Ottawa, 501 Smyth Road, Ottawa, ON K1H8L6, Canada; smart183@uottawa.ca

[6] Department of Medicine, Division of Medical Oncology, University of Ottawa, 501 Smyth Road, Ottawa, ON K1H8L6, Canada; ccanil@toh.ca

* Correspondence: lburgess@toh.ca

**Abstract:** Approximately 20% of renal cell carcinoma (RCC) is diagnosed because of paraneoplastic manifestations. RCC has been associated with a large variety of paraneoplastic syndromes (PNS), but it is rarely associated with PNS vasculitis. We present a case of a previously healthy male who presented with systemic vasculitis; bitemporal headaches, diplopia, polyarthritis, palpable purpura, tongue lesion, peri-orbital edema, scleritis, chondritis and constitutional symptoms. He was subsequently found to have oligometastatic RCC. Both his primary lesion and site of oligometastasis were treated with stereotactic radiotherapy (SBRT) and resulted in the resolution of his vasculitis, as well as sustained oncologic response. This is the first case to demonstrate that effective sustained treatment for PNS vasculitis due to oligometastatic RCC is possible with SBRT.

**Keywords:** stereotactic radiotherapy (SBRT); renal cell carcinoma (RCC); paraneoplastic vasculitis; paraneoplastic syndrome; oligometastases

## 1. Introduction

The classic triad in renal cell carcinoma (RCC) of flank pain, gross hematuria and a palpable mass is seen in approximately 10% of cases [1]. Secondary to improved diagnostic imaging, most RCCs are detected incidentally. However, approximately 20% of RCC patients are diagnosed as a result of paraneoplastic manifestations [2]. A wide range of paraneoplastic syndromes (PNS) are associated with RCC, but PNS vasculitis is very rare [3]. Herein, we review a case of a previously healthy man diagnosed with oligometastatic RCC following an atypical presentation of systemic vasculitis who demonstrated sustained and effective resolution of his vasculitis following treatment with stereotactic radiotherapy (SBRT).

## 2. Case

A 60-year-old male with a past medical history significant for asthma, allergic rhinitis, and dyslipidemia presented to the emergency department with a three-day history of binocular diplopia, right eye pain, diffuse joint pain and edema in the feet, ankles and wrists bilaterally, and a rash on the feet tracking up to the groin. Two months prior to this presentation, he developed persistent bi-frontal and bi-occipital headaches, fevers,

night sweats, and a swollen nose that failed two courses of outpatient antibiotics. Other outpatient investigations including a computed tomography (CT) head, CT venogram, and lumbar puncture; they were non-contributory.

He was found to have right peri-orbital edema and conjunctival injection, Figure 1A, with preserved extra-ocular movements and no ptosis. He had a circular 1 cm × 1 cm white lesion on his lateral right tongue, Figure 1B, edema/erythema of the nasal bone and upper cartilage, and cervical lymphadenopathy. He was noted to have bilateral synovitis of the metatarsophalangeal joints, ankles, elbows, wrists, and metacarpophalangeal joints. His purpuric rash was palpable, non-blanching, and extended from his lower extremities to his buttocks. The remainder of his exam was normal.

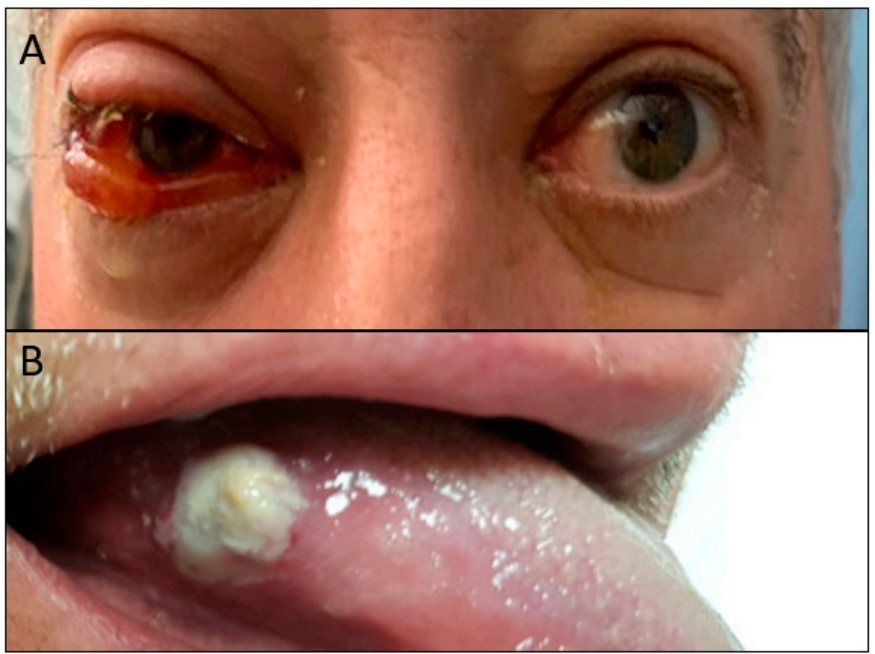

**Figure 1.** (**A**) Right peri-orbital edema and conjunctival injection at presentation; (**B**) The circular 1 cm × 1 cm white lesion on his lateral right tongue at presentation.

Initial investigations revealed a normocytic anemia (Hb 115 g/L) and neutrophilic leukocytosis (white blood cell count $20.2 \times 10^9$/L, neutrophils $17.3 \times 10^9$/L). Renal function was maintained; however, urine studies revealed hematuria and non-nephrotic proteinuria (2.48 g/day). Liver enzymes revealed a mixed transaminitis (AST 90 U/L, ALT 123 U/L, ALP 242 U/L, GGT 448 U/L). Inflammatory markers were elevated (erythrocyte sedimentation rate 122 mm/h, C-reactive protein > 190 mg/L). CT orbit revealed soft tissue thickening, fat stranding, and enhancement in the right pre-septal region. CT head and neck revealed bilateral prominent neck lymphadenopathy, more extensive on the right. CT abdomen and pelvis revealed a 1.9 cm indeterminate isodense to hypodense lesion in the mid to lower pole of the left kidney, Figure 2A. He was pan-cultured, started on piperacillin/tazobactam, and admitted to internal medicine. Infectious diseases and rheumatology were consulted. The differential remained broad and included infectious, autoimmune, and malignant aetiologies.

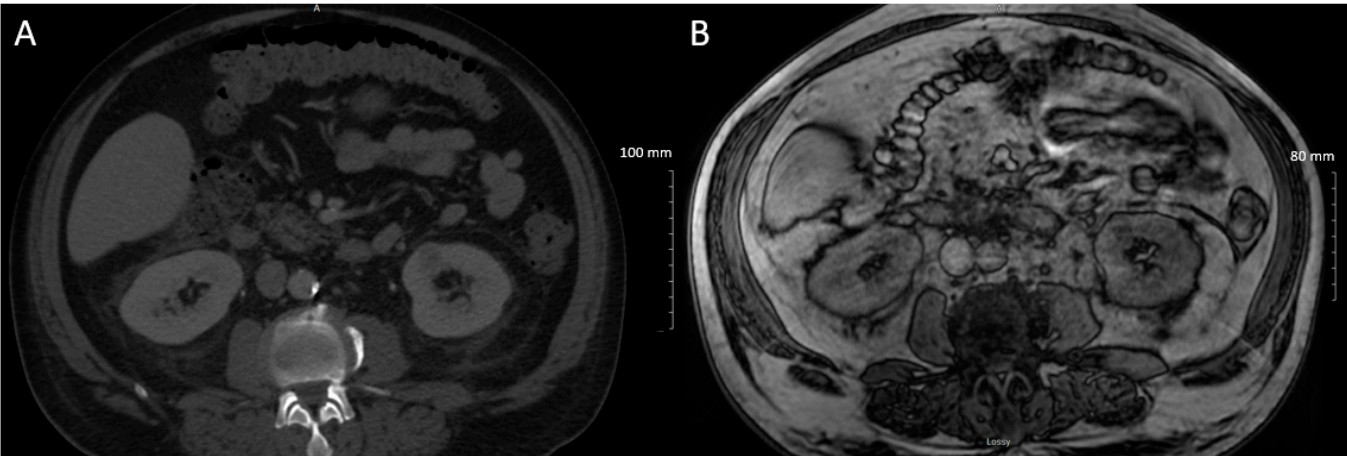

**Figure 2.** (**A**) Axial CT revealing indeterminate 18 mm left renal lesion; (**B**) Axial dynamic gradient echo sequence demonstrating 17 mm T2 hypointense left renal lesion in keeping with renal cell carcinoma.

An extensive infectious work-up returned negative; including blood cultures, urine cultures, stool cultures, stool ova and parasites, Chlamydia/Gonorrhea, and serologies for HIV, hepatitis B, hepatitis C, cytomegalovirus, Epstein-Barr virus, lyme, and syphilis. An autoimmune work-up revealed a positive RF (123 kIU/L), anti-CCP (250 U), weakly positive ANA (1:80, speckled), ENA positive for anti-Ro52/TRIM 21 (45), and IgG4 (2.97 g/L). IgE was elevated at 3101 μg/L. Double-strand DNA, complement, ANCA, immunoglobulins, cryoglobulins, APLA, ACE, CK, and triglycerides were negative. Investigations for hematologic malignancy were also negative and included a normal serum and urine protein electrophoresis, negative beta-2-microglobulin, and a negative skeletal survey. Hemolytic work-up was negative. With these results, an autoimmune etiology was strongly favoured, although symptomatology and serology were not consistent with a primary vasculitis; antibiotics were stopped and he was started on prednisone 60 mg PO daily. Of note, ophthalmology was consulted for his ocular symptoms, which was felt to be scleritis secondary to a systemic autoimmune disorder.

Biopsy of his tongue lesion was negative for fungal infection, malignancy, and IgG4. A skin biopsy was taken of his purpuric lower extremity rash. Pathology revealed leukocytoclastic vasculitis involving small vessels. Direct immunofluorescence biopsy was positive for IgM and C3 in dermal vessel walls, IgA negative.

An abdominal MRI was performed to follow up on the indeterminate left renal lesion seen on CT abdomen and pelvis. This revealed a 17 mm T2 hypointense enhancing nodule in the interpolar region of the left kidney, most likely representing a renal cell carcinoma (RCC), Figure 2B. Given his atypical symptomatology and investigations, rheumatology was highly suspicious of PNS vasculitis secondary to RCC. Urology was consulted and decided that this presumed RCC was unlikely to account for his constellation of symptoms, given its small size.

A CT chest for parenchymal and vascular assessment was performed and revealed a focal osteolytic lesion involving the body of the left scapula with suspected overlying cortical breach, Figure 3A.

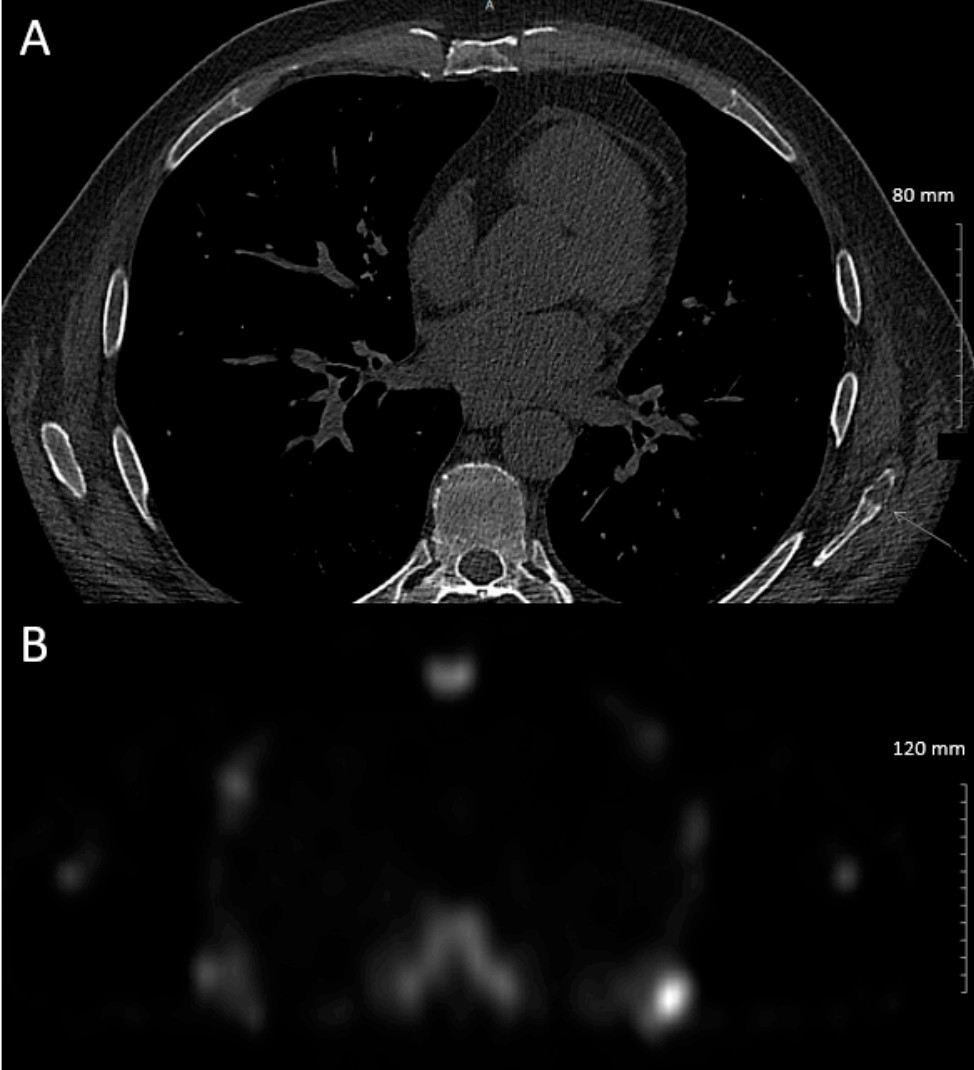

**Figure 3.** (**A**) Axial CT chest demonstrating osteolytic lesion at inferior tip of left scapula with suspected overlying cortical breach; (**B**) Bone scan revealing abnormal focal radiotracer uptake in the inferior tip of the left scapular body, correlating to abnormality seen on CT chest.

He improved significantly with steroids; his headaches, fevers, arthritis, edema, tongue lesion, peri-orbital edema, scleritis, chondritis, leukocytoclastic vasculitis and neck adenopathy improved and his leukocytosis trended down. He was discharged home on a tapering dose of steroids, hydroxychloroquine, a bone scan to follow up on left scapula lesion, and urology and rheumatology follow-up.

A bone scan confirmed focal radiotracer uptake in the left scapula, correlating to the lesion seen on CT, Figure 3. Biopsy was arranged and pathology revealed metastatic clear cell renal cell carcinoma. Staging scans did not reveal any further sites of metastases.

His case was discussed at multidisciplinary cancer conference rounds and the consensus was to treat both the renal primary and the scapular oligometastatic site with SBRT, rather than offer a nephrectomy or upfront systemic therapy, considering his vasculitis and steroids. He was treated with definitive SBRT; 30 Gy in three fractions to the left scapular metastasis and then 39 Gy in three fractions using CyberKnife, a radiosurgery platform, to the left renal primary. He tolerated radiotherapy well, except for a flare in scapular pain that subsequently resolved.

Five months after completion of his left scapular SBRT, with effective treatment of his underlying malignancy, he had been completely tapered off prednisone and hydroxychloroquine and his vasculitis was in remission. Improvement with oncologic therapy,

and without stronger immunosuppression typically required for primary vasculitis is suggestive of PNS vasculitis. Imaging with an MRI abdomen seven months post-SBRT revealed stable appearance of his left renal RCC, measuring 1.9 cm × 2.1 cm × 2.0 cm (AP × transverse × cranio-caudal), Figure 4A. This is in keeping with good response to radiotherapy. A bone scan one year post-SBRT demonstrated interval improvement of the left scapular bony lesion, Figure 4B. No further sites of metastases have been identified on restaging investigations and clinical assessment, 16 months post-SBRT.

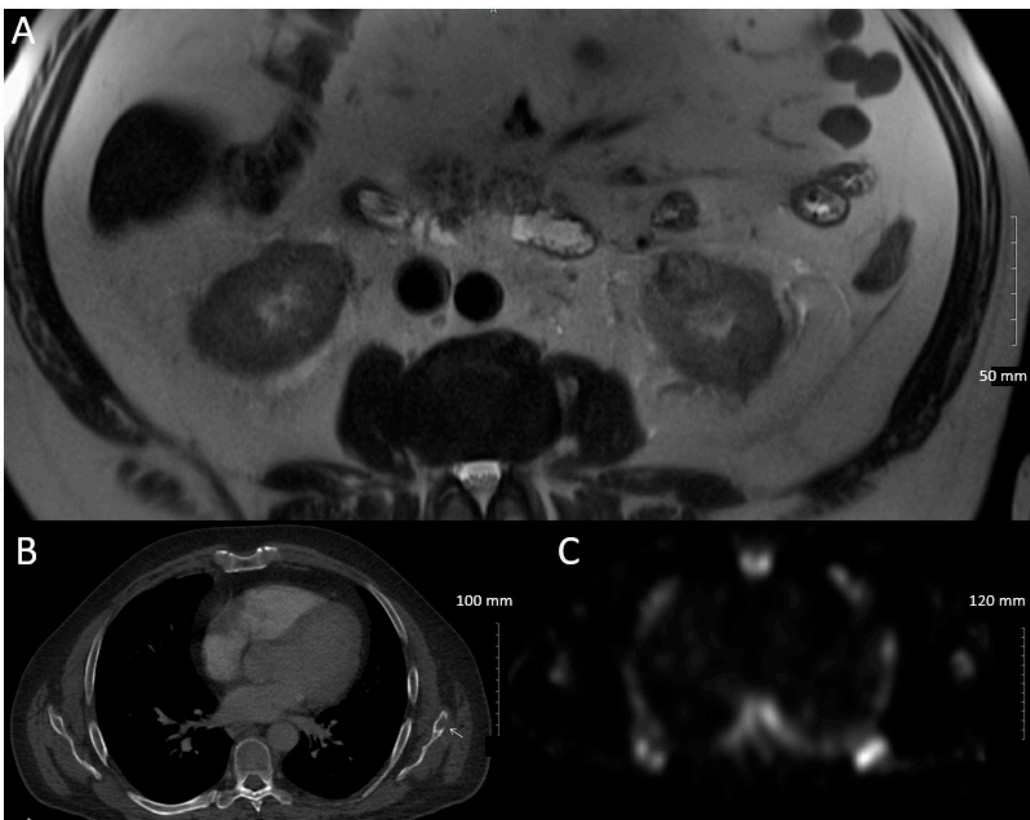

**Figure 4.** (**A**) Axial echo planar fast spin echo MR images showing stability of the left renal lesion that is heterogeneously T2 iso- and hypointense; (**B**) Bone scan showing stability of abnormal focal radiotracer uptake in the inferior tip of the left scapular body; (**C**) CT showing stability of lesion in the left scapula.

## 3. Discussion

Paraneoplastic syndromes (PNS) are a set of signs, symptoms and laboratory findings secondary to a malignant tumor that are neither associated with the primary site, metastasis, infection, nutritional deficiency, or treatment [4]. They are thought to arise secondary to release of tumor-associated proteins secreted by the tumor cells or by the immune system in response to the tumor [2]. Consequently, any part of the body may be involved.

While 20% of patients diagnosed with RCC present with PNS, another 10–40% develop PNS over the course of their disease [2]. There are 15 different types of PNS that have been described in the context of RCC, contributing to its historical reference as the "internist's tumor", including: constitutional symptoms, hypercalcemia, hypertension, polycythemia, hepatic dysfunction, hyperglycemia, amyloidosis, anemia, neuromyopathy, vasculopathy, and coagulopathy [5,6]. Hypercalcemia is the most common PNS in patients with RCC [5]. PNS manifestations are often assumed to be a harbinger of worse prognosis [2]. The Mayo Clinic performed a retrospective review of patients undergoing nephrectomy for localized RCC from 1990 and 2010 and found that patients presenting with PNS had worse disease

characteristics than those who were asymptomatic, leading to worse overall survival and cancer-specific survival [3].

PNS associated with small RCC tumors have rarely been reported [7], but this is seen in our patient with a very small primary and only one site of oligometastatic disease. As PNS mostly resolve with appropriate treatment of the underlying malignancy, the gold standard in the treatment of localized RCC PNS is surgical removal of the primary. Resolution with oncologic treatment is considered an important part in the confirmation that it is truly a paraneoplastic process [8], and persistence of PNS following appropriate oncologic treatment is considered a poor prognostic factor and can signal recurrence of malignancy [6].

Vasculitis represents a spectrum of syndromes with vascular inflammation and damage to blood vessels, and can be localized or systemic [9,10]. Vasculitis may be primary or secondary to infection, other autoimmune conditions or malignancy. The specific mechanism of PNS vasculitis is unknown, but it is posited that it may be the result of immune response or tumor-associated proteins [2], resulting in vasopermeability, vasodilation (appears as edema and erythema), dysfunction of endothelial cells, perivascular inflammatory cells, platelet aggregation and deposition of fibrin intravascularly [9]. Cutaneous PNS vasculitis is the most common manifestation [9]. Systemic PNS vasculitis, as seen with our patient, is rare.

It is hypothesized that PNS vasculitis secondary to any malignancy is from either the presence of tumor-associated proteins or the body's immune response to the presence of the tumor [2]. Regardless of the primary malignancy, it is associated with a worse oncologic prognosis, but PNS vasculitis is most commonly seen in hematological malignancies [8,9]; therefore, hematologic work-up was performed when PNS vasculitis was suspected. While PNS vasculitis is associated with worse oncologic prognosis, the vasculitis itself can be cured with effective oncologic treatment. This differs from primary vasculitis, which requires prolonged immunosuppression [8].

PNS vasculitis is more frequently associated with hematologic malignancies [8,9], with less common association with solid tumors [8,9]. As a result, most of the studies on PNS vasculitis in solid tumors are case reports or small series of patients [2,3,8,10–14]. PNS vasculitis in RCC has only been reported in a few case reports, where treatment of RCC and PNS vasculitis was with nephrectomy [2,8,10,15,16]. Hoag described two cases of patients who had leukocytoclastic vasculitis and temporal arteritis, respectively, who died while in hospital, and autopsy later confirmed RCC in both cases [7]. Curgunlu et al. presented a case of a man who presented with leukocytoclastic vasculitis and was found to have an RCC. He was treated with a nephron-sparing surgery, after which his vasculitic lesions disappeared [15]. There are no previous reports of the use of SBRT in sustained long-term resolution of PNS vasculitis in RCC. As our case illustrates, radiation with SBRT can be considered an alternative form of localized treatment with excellent results to date for our patient, given resolution of his vasculitis and radiological response with no evidence of progression.

## 4. Conclusions

In summary, we present a case of a paraneoplastic syndrome secondary to oligometastatic renal cell carcinoma with rheumatologic manifestations consistent with systemic vasculitis, including bitemporal headaches, diplopia, polyarthritis, palpable purpura, tongue lesion, peri-orbital edema, scleritis, chondritis and constitutional symptoms with sustained response following SBRT to both the primary and site of oligometastatic disease, without requiring the immunosuppression that would be required for primary vasculitis. PNS vasculitis is rarely seen in RCC, particularly in small RCC, only reported in case reports, and is typically treated with surgery. This case contributes to the limited literature available on PNS vasculitis in RCC and is the first to demonstrate that the use of stereotactic radiotherapy to both the site of the primary lesion and oligometastatic lesion results in the resolution of vasculitis and

sustained oncologic response. This case demonstrates that effective, sustained treatment for PNS vasculitis secondary to oligometastatic RCC is possible with stereotactic radiotherapy.

**Author Contributions:** Conceptualization, L.B. and R.M.; literature review, L.B., A.L.Z., S.M., J.Y.; writing—original draft preparation, L.B.; writing—review and editing, M.K., A.L.Z., K.L., D.H.S., J.Y., S.M., C.M., J.W., C.C., R.M.; supervision, R.M. All authors have read and agreed to the published version of the manuscript.

**Funding:** This research received no external funding.

**Institutional Review Board Statement:** Not applicable.

**Informed Consent Statement:** Written informed consent has been obtained from the patient to publish this paper.

**Conflicts of Interest:** C.M. reports personal fees from Amgen, Abbvie, Astellas, Bayer, Janssen, Sanofi, TerSera, Knight and Ferring. C.C. reports personal fees from BMS, Elsai, Pfizer, Ipsen, Sanofi Genzyme, Roche, Janssen, Amgen, Merck, Ferring, Astellas, as well as non-financial support from Janssen and Pfizer. The funders had no role in the design of the study; in the collection, analyses, or interpretation; in the writing of the manuscript, or in the decision to publish the results.

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
