# Peer review of "Stereotactic Radiotherapy in the Treatment of Paraneoplastic Vasculitis in Oligometastatic Renal Cell Carcinoma"

_curroncol, doi:10.3390/curroncol28030162_

Round 1

Reviewer 1 Report

This is an interesting case that is comprehensively reported.   I would suggest a few changes:

  1. The lesions are quite small;  it might be better to have one larger representative  before and after image of the renal and scapular lesions (i.e. just one axial view before and after and cropped to just the area of interest).
  2. The size of the renal lesion is stated as 1.7mm on line 106, elsewhere it is 17mm;   careful attention to make sure the reporting of measurements throughout is consistent is suggested
  3. Was there resolution of serologic markers of vasculitis after treatment as well?
  4. Given the unusual presentation, the authors should at least entertain the possibility that the patient may have both "ticks and fleas"  - I don't know that you can't definitively conclude this was PNS;  it is at least possible that it was a coincidental occurrence of vasculitis with incidental finding of renal cancer.     The concurrent treatment of the vasculitis with steroids and methotrexate (while clinically necessary) potentially obscures the contribution of the radiation treatment to the resolution of the symptoms.    Perhaps a reference that calibrates expectation of response to treatment in non PNS vasculitis?   Was resolution of symptoms quicker than one would expect with non PNS treated similarly - if yes, would strengthen the case that radiosurgery contributed to the outcome

Author Response

  1. The lesions are quite small;  it might be better to have one larger representative  before and after image of the renal and scapular lesions (i.e. just one axial view before and after and cropped to just the area of interest).

Thank you for the recommendation, we agree that larger axial images better convey the information and have removed sagittal images. We have now replaced our figures with those that are cropped highlighting the areas of interest.

  1. The size of the renal lesion is stated as 1.7mm on line 106, elsewhere it is 17mm;   careful attention to make sure the reporting of measurements throughout is consistent is suggested

Thank you for noting our error. We have changed this appropriately and have reviewed the remainder of the manuscript to ensure there are no other inconsistencies.

  1. Was there resolution of serologic markers of vasculitis after treatment as well?

One of the features that distinguishes this from a possible primary vasculitis or vasculitis from another cause (other than PNS) is that there were no serologic markers that supported a diagnosis of primary vasculitis. As such, they were never abnormal. We have explicitly stated this in the manuscript now for additional clarity.

  1. Given the unusual presentation, the authors should at least entertain the possibility that the patient may have both "ticks and fleas"  - I don't know that you can't definitively conclude this was PNS;  it is at least possible that it was a coincidental occurrence of vasculitis with incidental finding of renal cancer.     The concurrent treatment of the vasculitis with steroids and methotrexate (while clinically necessary) potentially obscures the contribution of the radiation treatment to the resolution of the symptoms.    Perhaps a reference that calibrates expectation of response to treatment in non PNS vasculitis?   Was resolution of symptoms quicker than one would expect with non PNS treated similarly - if yes, would strengthen the case that radiosurgery contributed to the outcome

We have more clearly stated in the manuscript how this case was a paraneoplastic vasculitis rather than both primary vasculitis and a malignancy. He did not require treatment with high-dose medications, only low-dose steroids and hydroxychloroquine. He did not require methotrexate or other stronger immunosuppression. He was cured of his vasculitis process with the treatment of his cancer. Had this been a primary vasculitis, when he was weaned off of these low-dose medications, his vasculitic process would have returned and would have required much stronger immunosuppression. We have now more clearly stated this in the manuscript and appreciate the comments.

Reviewer 2 Report

This exceptional clinical case is nicely presented and conveniently framed in a well documented discussion. No additional comments or suggestions from my part.

Author Response

This exceptional clinical case is nicely presented and conveniently framed in a well documented discussion. No additional comments or suggestions from my part.

We appreciate your feedback and enthusiasm for our manuscript

Reviewer 3 Report

Re: clinical case report

“Stereotactic Radiotherapy in the Treatment of Paraneoplastic Vasculitis in Oligometastatic Renal Cell Carcinoma” – Current Oncology

This manuscript describes a clinical case related to the possible benefits of the stereotactic radiotherapy for the treatment of a paraneoplastic syndrome (PNS), vasculitis, in a patient with oligometastatic renal cell carcinoma (RCC). This is a very interesting paper which addresses a rare but recognized PNS in RCC.

The paper is written well and present the case in a very professional way.

I have a few comments:

  • The authors should add a paragraph in the discussion section associated to the incidence, causes and prognosis of PNS vasculitis occurred in the other types of cancers.
  • There also few missing papers related to the topic (PMID: 32721112, 32128064, 33440621, and 8446279).
  • The time of the last follow-up must be clearly stated (including the findings).

Author Response

This manuscript describes a clinical case related to the possible benefits of the stereotactic radiotherapy for the treatment of a paraneoplastic syndrome (PNS), vasculitis, in a patient with oligometastatic renal cell carcinoma (RCC). This is a very interesting paper which addresses a rare but recognized PNS in RCC.

The paper is written well and present the case in a very professional way.

I have a few comments:

  • The authors should add a paragraph in the discussion section associated to the incidence, causes and prognosis of PNS vasculitis occurred in the other types of cancers.

We appreciate this suggestion and have now incorporated discussion about incidence, causes and prognosis of PNS vasculitis in other malignancies. We feel that this addition has strengthened our discussion and appreciate the suggestions

  • There also few missing papers related to the topic (PMID: 32721112, 32128064, 33440621, and 8446279).

We appreciate the additional references that have been suggested and have now incorporated them into the manuscript. We believe that this strengthens the manuscript overall and highlights the rarity of paraneoplastic vasculitis in renal cell carcinoma, without reports of treatment with stereotactic radiotherapy.

  • The time of the last follow-up must be clearly stated (including the findings).

We have tried to more clearly describe the timing of last follow up and have included more recent images (bone scan) to reflect the updated findings, since the original submission of the manuscript.